# Efficient Backpropagation with Variance-Controlled Adaptive Sampling

**Ziteng Wang, Jianfei Chen**[1]**, Jun Zhu**
Dept. of Comp. Sci. and Tech., Institute for AI, BNRist Center, THBI Lab,
Tsinghua-Bosch Joint ML Center, Tsinghua University
`wangzite23@mails.tsinghua.edu.cn`; `{jianfeic, dcszj}@tsinghua.edu.cn`

## Abstract

Sampling-based algorithms, which eliminate "unimportant" computations during forward and/or back propagation (BP), offer potential solutions to accelerate neural network training. However, since sampling introduces approximations to training, such algorithms may not consistently maintain accuracy across various tasks. In this work, we introduce a variance-controlled adaptive sampling (VCAS) method designed to accelerate BP. VCAS computes an unbiased stochastic gradient with fine-grained layerwise importance sampling in data dimension for activation gradient calculation and leverage score sampling in token dimension for weight gradient calculation. To preserve accuracy, we control the additional variance by learning the sample ratio jointly with model parameters during training. We assessed VCAS on multiple fine-tuning and pre-training tasks in both vision and natural language domains. On all the tasks, VCAS can preserve the original training loss trajectory and validation accuracy with an up to 73.87% FLOPs reduction of BP and 49.58% FLOPs reduction of the whole training process. The implementation is available at https://github.com/thu-ml/VCAS.

## 1 Introduction

Training neural networks can be computationally intensive. Contemporary networks typically employ stochastic gradient methods (Bottou et al., 2018) for training, which iteratively process batches of data to compute stochastic gradients through forward propagation (FP) and back propagation (BP) techniques (Rumelhart et al., 1986). FP+BP are costly, as they need to process every datum in the batch and every connection in the network, resulting in a multiplicative time complexity of batch size and model size. Such a time complexity becomes increasingly problematic in the era of big data and big models.

Data samples are not equally important. Some might be easy for the network to learn, while others might be extremely hard. Training can be accelerated by utilizing this disparity, focusing the available computational resources on more pivotal samples. At a high level, this can be achieved by further sampling the batch with higher keep probability of more important samples. The computational overhead is consequently diminished, in proportion to the quantity of retained samples. Various methods are proposed to assess the importance of samples, including meta-learning methods (Fan et al., 2017; Coleman et al., 2019; Mindermann et al., 2022), loss-based methods (Loshchilov & Hutter, 2015; Chang et al., 2017; Jiang et al., 2019; Ouyang et al., 2022),

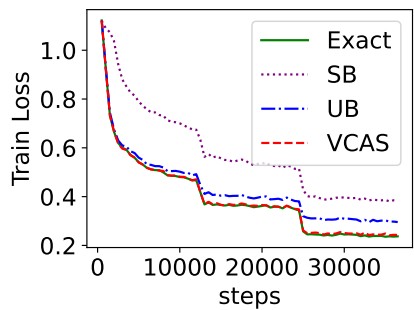

Figure 1: VCAS mirrors the convergence trajectory with exact training with FLOPs redution of 41.56%. Other methods like SB (Jiang et al., 2019) and UB (Katharopoulos & Fleuret, 2018) fail with a similar FLOPs reduction.

---

[1]Corresponding author.

and gradient norm based methods (Needell et al., 2014; Zhao & Zhang, 2015; Alain et al., 2015; Johnson & Guestrin, 2018; Katharopoulos & Fleuret, 2018).

While such methods seem promising, one core concern of sampling-based methods is their robustness. Misjudging the importance can hamper convergence, potentially leading to degraded accuracy and even longer training time than uniform sampling. Moreover, the optimal sample ratio is influenced by data distribution, which differs between tasks and is challenging to determine in advance. In general, there is a "no-free-lunch" phenomenon (Kaddour et al., 2023), where aggressive sampling often comes at the cost of reduced robustness.

In this work, we propose a robust variance-controlled adaptive sampling (VCAS) algorithm for deep learning under the stochastic optimization framework. VCAS computes a cost-effective approximated stochastic gradient (ASG) by partially conducting backpropagation for specific data and tokens. This ASG is unbiased, and we have developed an adaptive sampling method to meticulously control the variance of the ASG, aligning it with the original stochastic gradient's variance. Consequently, convergence remains largely unaffected, with our method mirroring the progression of exact algorithms, as delineated in Fig. 1.

Unlike previous methods, VCAS construct the ASG in a fine-grained manner. Rather than dropping samples one-time in a whole, VCAS gradually drops more samples when backpropagating from topmost to bottommost network layers, as the gradient getting sparser. Furthermore, VCAS also more aggressively drops data in finer granularity of tokens rather than samples when computing the weight gradients. VCAS can achieve smaller variance under a given computational budget compared to coarse grained sampling on the data dimension.

We evaluate VCAS on multiple finetuning and pre-training tasks of language models and vision transformers. VCAS can preserve the original training loss trajectory and the validation accuracy on all tasks, while adaptively determining the computational saving depending on the difficulty of the task. VCAS can reduce the computational cost of backpropagation by up to 73.87%, and reduce the overall training computation by up to 49.58%.

## 2 RELATED WORK

Methods focusing on the difference of data, known as online batch selection (Loshchilov & Hutter, 2015), can be mainly categorized into three classes: meta learning methods, loss based methods and gradient norm based methods. In this section we will discuss these three ways separately and briefly introduce other orthogonal efficient training methods.

**Meta Learning Methods.** Some works formulate data sampling into an optimization problem and train a separate meta predictor to solve it. Fan et al. (2017) use deep reinforcement learning to train an agent for data selection. Coleman et al. (2019) and Mindermann et al. (2022) train a separate cheaper model with similar architecture for guidance. However, training a meta predictor will introduce further overhead and it's a non-trivial learning task with more uncertainty introduced for weak theoretical guarantee.

**Loss Based Methods.** Loss is a natural indicator of the importance of different data. Loshchilov & Hutter (2015) maintains a history of losses and develops a sophisticated distribution based on the value or rank of loss. Jiang et al. (2019) and Ouyang et al. (2022) simplify it with sampling distribution proportion to the percentile of loss in the history. Chang et al. (2017) broadens the history to every datum and proposes to sample by the variance of prediction probability directly linked with previous losses. Dong et al. (2021) provides another method of minimizing the $L_2$ norm between the sampled loss and the exact counterpart. Shah et al. (2020) samples the smallest loss for robustness to outliers. Zhang et al. (2023) ensembles several loss methods with a preset sample ratio and varies the weights assigned to these methods adaptively. Simple and effective as they may be, the loss based methods are heuristic and always need a hyperparameter of sample ratio to tune for different tasks, violating the goal of efficient training.

**Gradient Norm Based Methods.** Previous works have proved that the optimal data sampling distribution for SGD is proportional to the gradient norm(Needell et al., 2014; Zhao & Zhang, 2015). But calculating the gradient norm is prohibitive since it needs a full process of backpropagation. To solve this problem, Alain et al. (2015) applies distributed training with many workers calculating

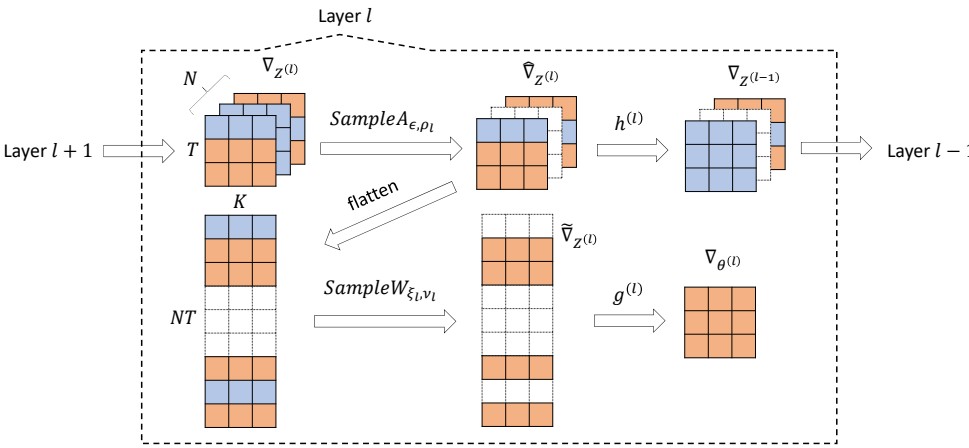

Figure 2: The computing diagram of backpropagation with VCAS in every layer. We use light blue squares to represent small gradient entries and orange for large ones. White squares are discarded by sampling. The upper line calculates activation gradient and the lower for weight gradient. Please refer to Sec. 4 for notations.

this importance score in parallel. Johnson & Guestrin (2018) uses a second-order approximation of gradient norm with history maintained. Closely related to our work, Katharopoulos & Fleuret (2018) develops a pure online algorithm by constructing an upper bound of gradient norm to sample with much cheaper computation. These methods are usually more expensive but have relatively strong theoretical guarantees. So we follow this way in our activation sampling.

**Orthogonal Efficient Training Methods.** Data pruning (Paul et al., 2021; Fayyaz et al., 2022) focuses on filtering less informative data before the whole training. Architecture pruning like layer dropping (Huang et al., 2016; Zhang & He, 2020) and token dropping (Hou et al., 2022; Yao et al., 2022; Li et al., 2022) modifies the architecture to make models faster to train with modest affect to performance. Mixed precision training and quantization (Micikevicius et al., 2018; Chen et al., 2021; Liu et al., 2022) change the training procedure to use low-precision in calculation for acceleration. Sparsity(Hoefler et al., 2021) focuses on pruning near-zero values in weights, activations, or gradients to achieve a low FLOPs(Raihan & Aamodt, 2020) and low memory footprint(Nikdan et al., 2023), yet is usually hard to bring a wall-clock time reduction like us due to the lack of hardware support(NVIDIA, 2021). All these works are orthogonal to our work since we focus on the computation approximation of a certain model architecture on a certain dataset with a certain training procedure to bring real training acceleration.

## 3 VARIANCE-CONTROLLED SAMPLING AS STOCHASTIC OPTIMIZATION

In this section, we present a high-level overview of our sampling algorithm as stochastic optimization. Consider the learning problem of a model $f(X; \theta)$ parameterized by $\theta$ on a dataset $\mathcal{D} = \{(X_i, y_i)\}_{i=1}^{|\mathcal{D}|}$ with a loss function $\ell(\cdot, \cdot)$. Define the learning objective as

$$\mathcal{L}(\theta) = \mathbb{E}_{\mathcal{B}}\left[\ell(f(X; \theta), y)\right], \tag{1}$$

where the expectation is taken over all possible batches $\mathcal{B} = (X, y)$ from $\mathcal{D}$. The model parameters can be learned by stochastic optimization algorithms (Bottou et al., 2018) with a stochastic gradient (SG) $g(\theta; \mathcal{B}) := \nabla_\theta \ell(f(X; \theta), y)$, which is an unbiased approximation of $\nabla_\theta \mathcal{L}(\theta)$.

However, computing the stochastic gradient can be still too expensive, since it requires the full forward and back propagation, which iterate over all model parameters and all data in the batch. We build a cheap stochastic approximation $g(\theta; \mathcal{B}, \epsilon)$ of the SG, which we refer as approximated stochastic gradient (ASG). ASG only computes the backpropagation partially, and is therefore cheaper than the SG. The randomness in the computing procedure of ASG is captured by $\epsilon$. We ensure that ASG is unbiased: $\mathbb{E}_\epsilon[g(\theta; \mathcal{B}, \epsilon)] = g(\theta; \mathcal{B})$.

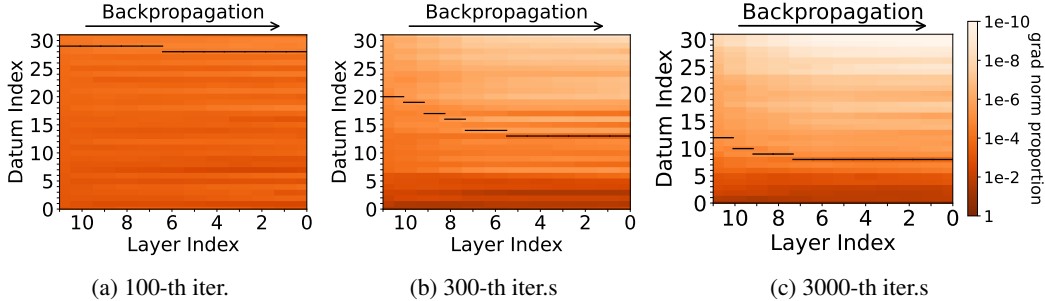

(a) 100-th iter.    (b) 300-th iter.s    (c) 3000-th iter.s

Figure 3: Gradient distribution over different layer and iterations of BERT-base finetuning on SST2 (6315 iterations in total). The normalized gradient norm of each datum is shown in the heatmaps. Black solid lines are the 95% percentile. Data above the lines are likely to be dicarded by VCAS.

With an unbiased SG, stochastic optimization algorithms are guaranteed to converge to a stationary point of Eq. (1), while the converge speed depends on the variance (cf. Bottou et al. (2018)). Therefore, if the variance of the ASG can be controlled to the similar variance level of SG, substituting the SG with ASG should have little impact to the convergence behavior. In fact, by the law of total variance (Chung, 2001), the variance of ASG can be decoupled as

$$\mathrm{Var}\left[g(\theta;\mathcal{B},\epsilon)\right] = \mathrm{Var}\left[g(\theta;\mathcal{B})\right] + \mathbb{E}_{\mathcal{B}}\left[\mathrm{Var}_\epsilon\left[g(\theta;\mathcal{B},\epsilon)\right]\right],$$

where the first term is the intrinsic variance of SG caused by subsampling batches from the dataset, and the second term is the additional variance incurred by ASG. In the subsequent sections, we will discuss our constructions of the ASG, which incurs negligible additional variance compared to SG.

## 4 FINE-GRAINED SAMPLING

Here we present variance-controlled adaptive sampling (VCAS), a specific construction of the ASG. We compute ASG by approximating the backpropagation in a fine-grained manner, and speed up matrix multiplications with importance sampling on the data dimension.

Assume a batch $X$ of shape $N \times T \times K$, where $N$ is the batch size, $T$ is the number of tokens of each datum, and $K$ is the dimensionality. For an $L$-layer network , the model $f(X;\theta)$ can be described by the following forward propagation procedure: $Z^{(0)} = X, Z^{(l)} = f^{(l)}\left(Z^{(l-1)};\theta^{(l)}\right), f(X;\theta) = Z^{(L)}$, where $Z^{(l)}$ and $\theta^{(l)}$ are the activation and parameters of the $l$-th layer, and $\theta = (\theta^{(l)})_{l=1}^L$. The SG can be computed by back-propagation in the following form: $\nabla_{Z^{(l-1)}} = h^{(l)}\left(\nabla_{Z^{(l)}}; Z^{(l-1)}, \theta^{(l)}\right), \nabla_{\theta^{(l)}} = g^{(l)}\left(\nabla_{Z^{(l)}}; Z^{(l-1)}, \theta^{(l)}\right)$, where $\nabla_{Z^{(l)}}$ and $\nabla_{\theta^{(l)}}$ denote the activation / weight gradient, $h^{(l)}$ and $g^{(l)}$ denote the function that calculates input / weight gradient of layer $l$ with the output gradient, layer input and weight. The SG $g(\theta;\mathcal{B}) = \left(\nabla_{\theta^{(l)}}\right)_{l=1}^L$.

As illustrated by Fig. 3, the activation gradients $\nabla_{Z^{(l)}}$ are sparse: the gradient $(\nabla_{Z^{(l)}})_i$ is close to zero for most sample $i$, except for a few important samples. Such sparsity becomes more prominent as backpropagating to lower layers and as the training progresses. To speed up computation, we add samplers in the backpropagation graph:

$$\hat{\nabla}_{Z^{(l)}} = \mathrm{SampleA}_{\epsilon,\rho_l}\left(\nabla_{Z^{(l)}}\right), \quad \nabla_{Z^{(l-1)}} = h^{(l)}\left(\hat{\nabla}_{Z^{(l)}}; Z^{(l-1)}, \theta^{(l)}\right),$$

$$\tilde{\nabla}_{Z^{(l)}} = \mathrm{SampleW}_{\xi_l,\nu_l}\left(\hat{\nabla}_{Z^{(l)}}, Z^{(l-1)}\right), \quad \nabla_{\theta^{(l)}} = g^{(l)}\left(\tilde{\nabla}_{Z^{(l)}}; Z^{(l-1)}, \theta^{(l)}\right). \quad (2)$$

The sampler $\mathrm{SampleA}_{\epsilon,\rho_l}(\cdot)$ randomly filter out unimportant data from the activation gradient, the keep ratio is $\rho_l$, with the randomness captured by $\epsilon$. The sampler is applied for each layer, so the activation gradient becomes increasingly sparse when backpropagating from the $L$-th layer to the first layer. The sampler $\mathrm{SampleW}_{\xi_l,\nu_l}(\cdot)$ filters (data, token) pairs specifically for weight gradient calculation, with a keep ratio $\nu_l$ and the randomness $\xi_l$. With these samplers, we only need to compute backpropagation for the retained data / token, so the computational cost is reduced. The sampling procedure is illustrated in Fig. 2, which constructs an unbiased ASG $g(\theta;\mathcal{B},\epsilon,\xi,\rho,\nu) = \left(\nabla_{\theta^{(l)}}\right)_{l=1}^L$, with $\nabla_{\theta^{(l)}}$ defined as Eq. (2), and $\xi = (\xi_l)_{l=1}^L, \rho = (\rho_l)_{l=1}^L, \nu = (\nu_l)_{l=1}^L$.

## 4.1 ACTIVATION GRADIENT

We apply unbiased low-variance approximation to the activation gradient to speed up subsequent computation. For an activation gradient tensor $G$ of shape $N \times T \times K$, we sample

$$\hat{G} = \text{SampleA}_{\epsilon,\rho}(G) = G \circ (m(\epsilon,\rho) \otimes \mathbf{1} \otimes \mathbf{1}),$$

where $\circ$ is element-wise product, and $\otimes$ is tensor outer product. The mask $m \in \mathbb{R}^N$ is a random Bernoulli vector: $m(\epsilon,\rho)_i = \text{Bern}(p_i;\epsilon)/p_i$, where $\sum_{i=1}^N p_i = N\rho$, and $\text{Bern}(p;\epsilon)$ denotes a Bernoulli random number generator with probability $p$ and randomness $\epsilon$. Since $\mathbb{E}[m(\epsilon,\rho)_i] = 1, \forall i$, the approximation is unbiased: $\mathbb{E}[\hat{G}] = G$. The sampler zeros out the gradient for all the data whose $m(\epsilon,\rho)_i = 0$. The amount of retained data is $N\rho$ in expectation. With the sampler, we only need to compute backpropagation for retained data, so the cost is $\rho$ times lower.

The variance of the approximation is $\text{Var}\left[\hat{G}\right] = \sum_{i=1}^N \frac{1-p_i}{p_i} \|G_i\|_F^2$, where we define the variance of a random tensor element-wise as $\text{Var}\left[\hat{G}\right] = \sum_{ijk} \text{Var}\left[\hat{G}_{ijk}\right]$, and $G_i$ denotes the $i$-th matrix of $G$ in the $N$ dimension. We compute the keep probability $(p_i)$ to minimize the variance, deriving a distribution proportional to the gradient norm of each datum: $p_i \propto \|G_i\|_F$. Minimizing the variance of the activation gradient not necessarily minimize the variance of ASG, which is the gradient of parameters. Nevertheless, this is a useful heuristic which empirically achieves low variance as is revealed by Katharopoulos & Fleuret (2018), and the ASG variance will be carefully controlled by our adaptive algorithm, as we shall see soon in Sec. 5.

## 4.2 WEIGHT GRADIENT

We can accelerate the computation of weight gradient for linear layers by sampling in both data and token dimensions. Consider the approximate back propagation of a linear layer $Z^{(l)} = Z^{(l-1)}\theta^{(l)\top}$:

$$\hat{\nabla}_{Z^{(l)}} = \text{SampleA}_{\epsilon,\rho_l}\left(\nabla_{Z^{(l)}}\right), \quad \tilde{\nabla}_{Z^{(l)}} = \text{SampleW}_{\xi_l,\nu_l}\left(\hat{\nabla}_{Z^{(l)}}, Z^{(l-1)}\right), \quad \nabla_{\theta^{(l)}} = \tilde{\nabla}_{Z^{(l)}}^\top Z^{(l-1)}$$

in matrix form, where we reshape the activation/gradients to $NT \times K$, and $\hat{\nabla}_{Z^{(l)}}$ is already a sampled matrix with only $NT\rho_l$ non-zero rows in expectation. However, $\hat{\nabla}_{Z^{(l)}}$ is only sampled in the data dimension. In fact, even $(\hat{\nabla}_{Z^{(l)}})_i$ is retained for some datum $i$, it might still have some rows (i.e., tokens) which are close to zero. We can further sample

$$\tilde{\nabla}_{Z^{(l)}} = \text{SampleW}_{\xi_l,\nu_l}\left(\hat{\nabla}_{Z^{(l)}}, Z^{(l-1)}\right) = \hat{\nabla}_{Z^{(l)}} \circ (m(\xi,\nu)^\top \mathbf{1}),$$

where the mask $m \in \mathbb{R}^{NL}$ is a random Bernoulli vector, and $\mathbf{1}$ is an all-one vector: $m(\xi,\nu)_i = \text{Bern}(q_i;\epsilon)/q_i$, where $\sum_{i=1}^{NT} q_i = NT\rho_l\nu_l$. The variance is

$$\text{Var}\left[\tilde{\nabla}_{\theta^{(l)}}\right] = \sum_{i=1}^{NT} \frac{1-q_i}{q_i} \left\|\hat{\nabla}_{Z^{(l)}_i}\right\|_2^2 \left\|Z_i^{(l-1)}\right\|_2^2. \tag{3}$$

The minimal variance solution is $q_i \propto \left\|\hat{\nabla}_{Z^{(l)}_i}\right\|_2 \left\|Z_i^{(l-1)}\right\|_2$. This sampling method is also known as leverage score sampling in randomized numerical linear algebra (Drineas & Mahoney, 2018).

## 5 ADAPTING SAMPLE RATIOS

The question remained is how to set the sample ratios $(\rho_l)_{l=1}^L$ and $(\nu_l)_{l=1}^L$. There is a tradeoff: lowering the sample ratio reduces the computational cost, but increases the variance. As discussed in Sec. 3, this ratio should be set to ensure that the additional variance of ASG is marginal compared to the original variance of SG. Adapting the sample ratio is nontrivial since the gradient sparsity pattern vary across layers and vary over time during training. In this section, we present an adaptation algorithm to control the variance during the entire training trajectory.

First, we introduce a single hyperparameter $s$ to control the sample ratios $(\rho_l)_{l=1}^L$ for all layers. Intuitively, when the gradient norm $(\|G_i\|_F)_{i=1}^N$ becomes sparser, we can more aggressively utilize

smaller keep ratio $\rho_l$ to maximize speedup. Therefore, we compute $\rho_l$ based on the sparsity $p_l$ of the gradient norm sequence:

$$p_l(s) = \min\{n/N | \sum_{i=1}^{n} \|G_i\|_F \geq s \sum_{i=1}^{N} \|G_i\|_F\}, \quad \rho_l(s) = \max_{j \leq l} p_j(s) \tag{4}$$

where $s \in [0, 1]$ is a hyperparameter on how much gradient norm is preserved. It's shown in Fig. 3 that gradient norm grows sparser with layer, yielding a descending trend of $p_l$ for $l$ from $L$ to 1. Thus it's reasonable to construct a monotone increasing sequence of $\{\rho_l\}_{l=1}^{L}$ based on $\{p_l\}_{l=1}^{L}$.

By law of total variance, we can decompose the variance of ASG as

$$\text{Var}\left[g(\theta; \mathcal{B}, \epsilon, \xi, \rho, \nu)\right] = \text{Var}\left[g(\theta; \mathcal{B})\right] + \mathbb{E}_{\mathcal{B}}[\text{Var}_\epsilon\left[g(\theta; \mathcal{B}, \epsilon, \rho(s))\right]] + \mathbb{E}_{\mathcal{B}, \epsilon}[\text{Var}_\xi\left[g(\theta; \mathcal{B}, \epsilon, \xi, \rho, \nu)\right]],$$

where we write $g(\theta; \mathcal{B}, \epsilon, \rho) := \mathbb{E}_\xi[g(\theta; \mathcal{B}, \epsilon, \xi, \rho, \nu)]$ to be the ASG without the sampler for weight gradient computation. The three variance terms are the SG variance, the variance introduced by approximately computing activation gradient, and the variance introduced by approximately computing weight gradient, respectively. Our algorithm adaptively tunes $s$ and $\nu$ during train to control the last two variance terms to be fractional comparing to the first variance term.

**Controlling** $\mathbb{E}_{\mathcal{B}}[\text{Var}_\epsilon\left[g(\theta; \mathcal{B}, \epsilon, \rho(s))\right]]$**:** We adopt a zeroth order method to adapt the hyperparameter $s$ to keep $\mathbb{E}_{\mathcal{B}}[\text{Var}_\epsilon\left[g(\theta; \mathcal{B}, \epsilon, \rho(s))\right]] = \tau_{act}\text{Var}\left[g(\theta; \mathcal{B})\right]$, where $\tau_{act} \ll 1$ is a small constant. That is, the additional variance raised by approximately computing activation gradient is only $\tau_{act}$ times the SG variance itself. Since larger $s$ increases the keep ratio and decreases the variance, we adopt the update:

$$s \leftarrow s + \alpha \, \text{sign}\left(\mathbb{E}_{\mathcal{B}}[\text{Var}_\epsilon\left[g(\theta; \mathcal{B}, \epsilon, \rho(s))\right]] - \tau_{act}\text{Var}\left[g(\theta; \mathcal{B})\right]\right), \tag{5}$$

where $\text{sign}(x) = +1$ when $x \geq 0$ and $\text{sign}(x) = -1$ when $x < 0$, and $\alpha$ is a step size. We approximate the expectation and variance with empirical ones with $M$ Monte Carlo repetitions. Therefore, each update requires $O(M^2)$ FP+BPs, and we run the update every $F$ SGD iterations, where $F \gg M^2$.

**Controlling** $\mathbb{E}_{\mathcal{B}, \epsilon}[\text{Var}_\xi\left[g(\theta; \mathcal{B}, \epsilon, \xi, \rho, \nu)\right]]$**:** As the variance sums up for each parameter $\theta^{(l)}$, we can further decompose the variance as

$$\mathbb{E}_{\mathcal{B}, \epsilon}[\text{Var}_\xi\left[g(\theta; \mathcal{B}, \epsilon, \xi, \rho, \nu)\right]] = \sum_{l=1}^{L} \mathbb{E}_{\mathcal{B}, \epsilon}\left[\text{Var}_\xi\left[g^{(l)}(\theta; \mathcal{B}, \epsilon, \xi_l, \rho, \nu_l)\right]\right], \tag{6}$$

where $g^{(l)}$ is the gradient of the $l$-th layer (i.e., $\nabla_{\theta^{(l)}}$). We control the variance of each layer separately to keep $\mathbb{E}_{\mathcal{B}, \epsilon}\left[\text{Var}_\xi\left[g^{(l)}(\theta; \mathcal{B}, \epsilon, \xi_l, \rho, \nu_l)\right]\right] = \tau_w\text{Var}\left[g^{(l)}(\theta; \mathcal{B})\right]$. Again, this is achieved by a zeroth-order algorithm:

$$\nu_l \leftarrow \nu_l \beta^{\text{sign}\left(\mathbb{E}_{\mathcal{B}, \epsilon}\left[\text{Var}_\xi\left[g^{(l)}(\theta; \mathcal{B}, \epsilon, \xi_l, \rho, \nu_l)\right]\right] - \tau_w\text{Var}\left[g^{(l)}(\theta; \mathcal{B})\right]\right)}, \tag{7}$$

where $\text{Var}_\xi\left[g^{(l)}\right]$ can be computed analytically by Eq. 3, and $\beta$ is a multiplier.

Now we are fully prepared to present the whole picture of VCAS in Alg. 1. Please refer to Appendix. D for more details about the algorithm.

## 6 EXPERIMENTS

### 6.1 TRAINING FLOPs REDUCTION

We assessed VCAS on multiple fine-tuning and pre-training tasks in both vision and natural language domains. We compare our algorithm with the exact training and two previous works in BP sampling: a loss based method SB(selective backprop) in Johnson & Guestrin (2018) and a gradient norm based method UB(upper bound) in Katharopoulos & Fleuret (2018). We choose these two methods since they are entirely online and need little modification to the original training pipeline like us. The results are shown in Tab. 1. All results are the average of 3 different seeds except for BERT-base pretraining and ViT finetuning on ImageNet-1k which we use 1.

---

**Algorithm 1** Variance controlled adaptive sampling(VCAS) for backpropagation

---

**Require:** update frequency $F$, Monte-Carlo repetition number $M$, variance tolerant ratio for activation $\tau_{act}$, for weight $\tau_w$, $s$ step size $\alpha$, weight ratio multiplier $\beta$

$s \leftarrow 1$, activation sample ratio schedule $\{\rho_l\}_{l=1}^L \leftarrow \mathbf{1}$, weight sample ratios $\{\nu_l\}_{l=1}^L \leftarrow \mathbf{1}$
$t \leftarrow 0$
**while** not converge **do**
    **if** $t \mod F = 0$ **then**
        **for** $i$ in $1, \ldots, M$ **do**
            $(X_i, y_i) \leftarrow$ batch selected randomly
            SGD gradient $G_{s,i} \leftarrow$ exact backward using $(X_i, y_i)$
            **for** $j$ in $1, \ldots, M$ **do**
                activation gradient $G_{act,i,j} \leftarrow$ backward using $(X_i, y_i)$ with SampleA only
                calculate weight variance $V_{w,i,j}$ analytically with Eq. 3 and Eq. 6
            **end for**
        **end for**
        SGD variance $V_s \leftarrow \frac{1}{M-1} \sum_{i=1}^M \left\| G_{s,i} - \frac{1}{M} \sum_{i=1}^M G_{s,i} \right\|_F^2$
        activation variance $V_{act} \leftarrow \frac{1}{M} \sum_{i=1}^M \left( \frac{1}{M} \sum_{j=1}^M \| G_{act,i,j} - G_{s,i} \|_F^2 \right)$
        weight variance $V_w \leftarrow \frac{1}{M} \sum_{i=1}^M \left( \frac{1}{M} \sum_{j=1}^M V_{w,i,j} \right)$
        update $s$ with $V_{act}$ and $V_s$ according to Eq. 5
        update $\{\rho_l\}_{l=1}^L$ with new $s$ according to Eq. 4
        update $\{\nu_l\}_{l=1}^L$ with $V_w$ and $V_s$ according to Eq. 7
    **end if**
    backward with SampleA and SampleW
    $t \leftarrow t + 1$
**end while**

---

Note that to avoid falling into the pitfall of unfair comparison with baseline which is not tuned under efficient settings as is pointed out by Dehghani et al. (2021) and Kaddour et al. (2023), for all these experiments we use the same conservative setting of $\tau_{act} = \tau_w = 0.025, \alpha = 0.01, \beta = 0.95, M = 2$. We preset all these values heuristically without any tuning or prior knowledge. The only hyperpamater we modified among different tasks is the variance calculation frequency $F$, which can be defined easily according to the total training steps.

In fact, all the hyperparameters introduced by VCAS have explicit meanings and are insensitive. We show experimentally that though extra tuning may achieve a slightly better result, overall VCAS is robust to these hyperparameters with reasonable values. Please refer to Appendix. A for details about ablation studies on these insensitive hyperparameters.

For SB and UB, we both adopt a sample ratio of 1/3, since it's the recommended setting in the original papers and it can achieve a FLOPs reduction of $1 - (1 + 2 * 1/3)/3 = 44.44\%$ which is close to the results we get in most tasks. An exception is BERT-base pretraining task where we find the FLOPs reduction achievable is low so we manually set the sample ratio of SB and UB to get the same FLOPs reduction as VCAS, so that they can still give a decent result. Nevertheless we are indeed favoring these methods by helping them to define a reasonable sample ratio, which can not be done themselves.

From the table we can see that overall VCAS is better than SB and UB with the least impact on final train loss and final evaluation accuracy. With FLOPs reduction of up to 49.58%, VCAS can still achieve nearly the same results with the exact counterpart.

## 6.2 WALL-CLOCK TIME REDUCTION

We record the wall-clock time of BERT-large finetuning on MNLI and ViT-large finetuning on ImageNet-1k with NVIDIA 3090Ti, the results are depicted in Tab. 2 and Tab. 3.

From these tables, we can find that VCAS can translate FLOPs reduction into wall-clock time reduction as effectively as simpler online batch sampling methods like UB and SB that drop part of

Table 1: Comparison of VCAS with other methods. Data format is *Final Train Loss / Final Eval Acc.(%)* for exact, SB and UB, and *Final Train Loss / Final Eval Acc.(%) / FLOPs reduction ratio(%)* for VCAS. The FLOPs reduction of SB and UB is 21.58% for BERT pretraining and 44.44% for other tasks. VCAS's FLOPs take account of the adaptation overhead. For BERT pretraining, accuracy=average performance on GLUE. Bold indicates the best result of each metric except for exact. Underline means Eval Acc less than 0.1% off the exact training.

| Task | Dataset | exact | SB | UB | VCAS |
|---|---|---|---|---|---|
| BERT-base pretraining | C4 | 2.099 / 78.37 | 2.133 / 77.53 | **2.106** / 77.96 | 2.134 / **78.36** / **21.58** |
| BERT-base finetuning | MNLI-m | 0.2372 / 84.33 | 0.3833 / 83.71 | 0.2957 / 83.82 | **0.2428** / **84.23** / 41.56 |
| | QQP | 0.1143 / 91.00 | 0.1441 / 90.76 | 0.1964 / 89.53 | **0.1189** / **90.92** / **47.10** |
| | QNLI | 0.1014 / 91.67 | 0.2017 / 90.58 | 0.1441 / 91.23 | **0.1056** / **91.29** / **44.45** |
| | SST-2 | 0.0559 / 92.59 | 0.0727 / 92.63 | 0.0743 / 92.82 | **0.0600** / **93.04** / **48.28** |
| BERT-large finetuning | MNLI-m | 0.1439 / 86.58 | 0.2492 / 85.18 | 0.2266 / 86.09 | **0.1619** / **86.63** / 44.17 |
| | QQP | 0.0885 / 91.64 | 0.1308 / 91.20 | 0.1751 / 90.51 | **0.0962** / **91.57** / **49.50** |
| | QNLI | 0.0877 / 92.02 | 0.1436 / 91.50 | 0.1325 / 91.98 | **0.0640** / **92.15** / **46.19** |
| | SST-2 | 0.0537 / 93.60 | 0.1136 / 91.81 | 0.0838 / 93.40 | **0.0593** / **93.67** / **49.24** |
| ViT-base finetuning | CIFAR10 | 0.1868 / 98.92 | 0.2367 / 98.82 | 0.1923 / **98.94** | **0.1873** / 98.90 / **45.90** |
| | CIFAR100 | 0.8760 / 91.19 | 2.248 / 89.60 | 1.175 / 89.68 | **0.8811** / **91.08** / 29.32 |
| | ImageNet-1k | 0.6032 / 82.27 | 0.6533 / 82.09 | 0.6109 / **82.28** | **0.6089** / 82.27 / **45.29** |
| ViT-large finetuning | CIFAR10 | 0.1359 / 99.24 | 0.1439 / 99.21 | **0.1378** / 99.17 | 0.1393 / **99.28** / **48.37** |
| | CIFAR100 | 0.4590 / 93.56 | 0.5983 / 93.07 | 0.5170 / 93.36 | **0.4649** / **93.64** / 38.67 |
| | ImageNet-1k | 0.4135 / 82.04 | 0.4637 / 82.21 | 0.4242 / 82.21 | **0.4228** / **82.27** / **49.58** |

Table 2: Wall-clock time of BERT-large finetuning on MNLI.

| Method | Train Loss | Eval Acc.(%) | Wall-clock Time(h) | FLOPs↓(%) | Time↓(%) |
|---|---|---|---|---|---|
| exact | 0.1439 | 86.58 | 5.478 | - | - |
| SB | 0.2492 | 85.18 | 4.320 | **44.44** | 21.14 |
| UB | 0.2266 | 86.09 | **4.266** | **44.44** | **22.12** |
| VCAS | **0.1619** | **86.63** | 4.437 | 44.17 | 19.00 |

Table 3: Wall-clock time of ViT-large finetuning on ImageNet-1k.

| Method | Train Loss | Eval Acc.(%) | Wall-clock Time(h) | FLOPs↓(%) | Time↓(%) |
|---|---|---|---|---|---|
| exact | 0.4135 | 82.04 | 52.29 | - | - |
| SB | 0.4637 | 82.21 | 42.56 | 44.44 | 18.61 |
| UB | 0.4242 | 82.21 | 41.92 | 44.44 | 19.83 |
| VCAS | **0.4228** | **82.27** | **41.28** | **49.58** | **21.06** |

data one-time in a whole, while enjoying mirrored performance with the exact training under theoretical guarantee.

The success of VCAS comes in two ways. One is the fine-grained sampling strategy that samples activation and weight jointly, which enables us to achieve much lower FLOPs given the variance budget. The other is the variance controlled framework combined with the self-adaptation algorithm, with which we are able to learn the proper sample ratios of different training phases. In the following two subsections, we will experimentally show the effectiveness of these two folds.

### 6.3 EFFECTIVENESS OF FINE-GRAINED SAMPLING

We compare VCAS that samples activation and weight jointly with strategies that solely sampling activation or weight. Specifically, we keep an equal extra variance for BERT-base finetuning on MNLI. We set $\tau_{act} = \tau_w = 0.025$ for VCAS, $\tau_{act} = 0.05$ for activation sampling only and $\tau_w = 0.05$ for weight sampling only. We find that under the preliminary that $\tau_{act}, \tau_w \ll 1$, the results

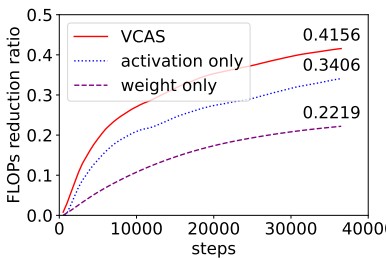
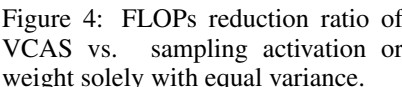

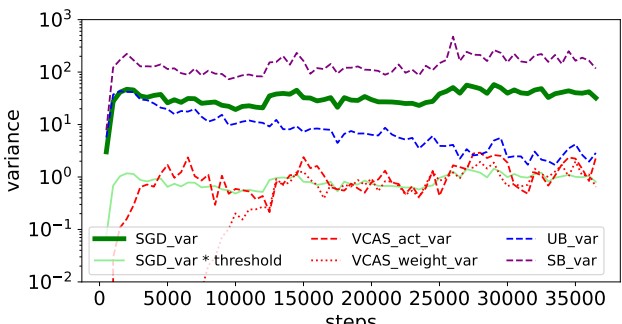

Figure 4: FLOPs reduction ratio of VCAS vs. sampling activation or weight solely with equal variance.

Figure 5: Gradient variance of different methods.

of these sampling strategies show no significant difference due to controlled variance. While as is shown in Fig. 4, VCAS can achieve a much greater FLOPs reduction with the same total variance introduced. It's reasonable since we can utilize more sparsity in both data and token dimensions with a fine-grained sampling strategy of VCAS.

### 6.4 EFFECTIVENESS OF VARIANCE CONTROL AND SELF-ADAPTATION

In Fig. 5 we plot the variance of different methods during training process of BERT-base finetuning on MNLI. We can find that VCAS is able to control the extra sampling variance introduced to our preset threshold, while for other variance-unaware algorithms like UB and SB, the extra variance is out of control with a similar FLOPs reduction.

With carefully controlled variance, a similar convergence with exact training is guaranteed as we mentioned in the introduction. As is depicted in Fig. 1 and Fig. 6 for BERT-base finetuning on MNLI, VCAS shares nearly the same convergence trajectory with the exact training with reduced FLOPs, while UB converges slightly slower due to uncontrolled variance, and SB converges in an entirely different trajectory with variance introduced far larger than exact.

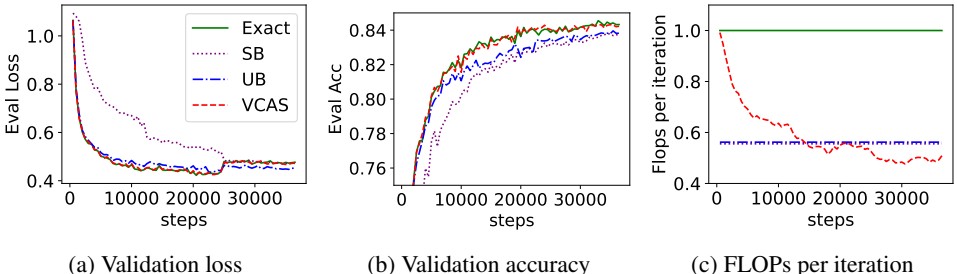

(a) Validation loss      (b) Validation accuracy      (c) FLOPs per iteration

Figure 6: Convergence comparison of different sampling methods. FLOPs is normalized by exact training.

## 7 CONCLUSION

We propose VCAS, a robust sampling method for back propagation with controlled variance and self-adaptive sample ratios. VCAS computes an approximate stochastic gradient by applying fine-grained sampling to gradually remove samples and tokens during backpropagation. VCAS enjoys similar variance, convergence trajectory, and final accuracy with exact back propagation, while reduces the training cost by up to 49.58%.

ACKNOWLEDGEMENTS

The authors would like to thank Bingrui Li and Weiyu Huang for their valuable discussions and help on algorithm design and implementation details. This work was supported by the National Key Research and Development Program of China (No. 2021ZD0110502), NSFC Projects (Nos. 62376131, 62061136001, 62106123, 62076147, U19A2081, 61972224), Tsinghua Institute for Guo Qiang, and the High Performance Computing Center, Tsinghua University. J.Z is also supported by the XPlorer Prize.

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

# A    ABLATION ON HYPERPARAMETERS

There are a few hyperparameters in our self-adaptation algorithm, but all of them have explicit meaning. In this section we show that though extra tuning of these hyperparameters may achieve a slightly better result, overall VCAS is robust to these hyperparameters with reasonable values. We conduct ablation experiments on two tasks: BERT-base finetuning on SST-2 and MNLI. All the results are averaged over 3 different seeds.

## A.1    ACTIVATION AND WEIGHT VARIANCE THRESHOLDS $\tau_{act}, \tau_w$

The main hyperparameters in VCAS is the variance thresholds of activation $\tau_{act}$ and weight $\tau_w$. For these two thresholds, how to split total variance among them is a big problem with optimal solution differing across models and tasks. So without prior knowledge introduced, we compromise by keeping $\tau_{act} = \tau_w = \tau \ll 1$.

We further conduct an ablation on $\tau$ from $0.01$ to $0.5$ as is shown in Tab. 4 for SST-2 and Tab. 5 for MNLI. From the results we can find that a satisfactory outcome is assured regardless of the specific value of $\tau$ provided that $\tau \ll 1$, which proves the robustness of VCAS.

Table 4: Ablation on different variance thresholds $\tau$ of BERT-base finetuning on SST-2

| $\tau$ | 0(exact) | 0.01 | 0.025 | 0.05 | 0.1 | 0.25 | 0.5 |
|---|---|---|---|---|---|---|---|
| Final Train Loss | 0.0559 | 0.0586 | 0.0600 | 0.0625 | 0.0642 | 0.0705 | 0.0761 |
| Final Eval Acc(%) | 92.59 | 93.07 | 93.04 | 93.25 | 92.81 | 92.79 | 92.18 |
| FLOPs reduction(%) | - | 45.92 | 48.28 | 49.82 | 50.05 | 51.57 | 52.71 |

Table 5: Ablation on different variance thresholds $\tau$ of BERT-base finetuning on MNLI

| $\tau$ | 0(exact) | 0.01 | 0.025 | 0.05 | 0.1 | 0.25 | 0.5 |
|---|---|---|---|---|---|---|---|
| Final Train Loss | 0.2372 | 0.2388 | 0.2428 | 0.2459 | 0.2552 | 0.2684 | 0.2805 |
| Final Eval Acc(%) | 84.33 | 84.31 | 84.23 | 84.33 | 84.07 | 84.13 | 84.08 |
| FLOPs reduction(%) | - | 38.59 | 41.56 | 43.49 | 45.37 | 47.53 | 48.92 |

## A.2    MONTE-CARLO REPETITIONS $M$

To calculate variances, VCAS introduces an overhead of extra iterations quadratic with Monte-Carlo repetitions $M$.

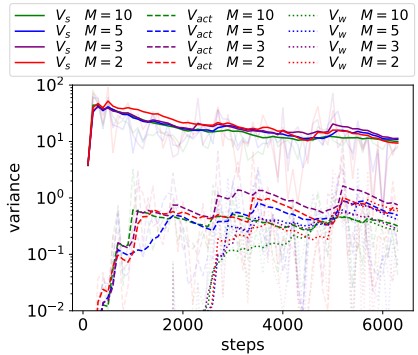 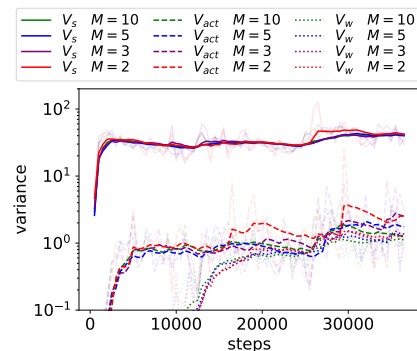

Figure 7: Variance calculated with different Monte-Carlo samples $M$ of BERT-base finetuning on SST-2.

Figure 8: Variance calculated with different Monte-Carlo samples $M$ of BERT-base finetuning on MNLI.

Obviously bigger $M$ will bring more precise empirical variance, yet the cost is prohibitive.

We experiment on different $M$ from 2 to 10 and find no significant difference in the empirical variance as is shown in Fig. 7 for SST-2 and Fig. 8 for MNLI. Therefore, we adopted the setting of $M = 2$, with which we only need to perform 6 extra iterations that is negligible if the variance calculation frequency is large enough like 100 in SST-2 and 500 in MNLI.

### A.3 VARIANCE CALCULATION FREQUENCY $F$

Similar to $M$, the variance calculation frequency $F$ is also a trade-off between better empirical approximation and less overhead introduced. We experimented on $F = 50, 100, 200, 500, 1000$ in Tab. 6 for SST-2 and Tab. 7 for MNLI. We can see that although as $F$ grows larger the overhead of VCAS is gradually relieved, with a too large $F$, like $F = 1000$ in SST-2 that leads to only 6 times of self-adaptation update, the sample ratio schedule is not fully explored and the final FLOPs reduction is even smaller. Therefore, for all these tasks we set $F$ to be at least $1/50$ of total training steps and no more than 500 due to slight marginal gains.

Table 6: Ablation on different adaptation frequency $F$ of BERT-base finetuning on SST-2, the number of training steps is 6315.

| $F$ | 0(exact) | 50 | 100 | 200 | 500 | 1000 |
|---|---|---|---|---|---|---|
| Final Train Loss | 0.0559 | 0.0589 | 0.0600 | 0.0587 | 0.0577 | 0.0562 |
| Final Eval Acc(%) | 92.59 | 92.71 | 93.04 | 92.56 | 93.15 | 93.19 |
| FLOPs reduction(%) | - | 47.33 | 48.28 | 46.06 | 39.43 | 31.03 |

Table 7: Ablation on different adaptation frequency $F$ of BERT-base finetuning on MNLI, the number of training steps is 36816.

| $F$ | 0(exact) | 50 | 100 | 200 | 500 | 1000 |
|---|---|---|---|---|---|---|
| Final Train Loss | 0.2372 | 0.2460 | 0.2461 | 0.2440 | 0.2428 | 0.2428 |
| Final Eval Acc(%) | 84.33 | 84.20 | 84.23 | 84.12 | 84.23 | 84.21 |
| FLOPs reduction(%) | - | 35.16 | 39.58 | 41.31 | 41.56 | 39.43 |

### A.4 $s$ UPDATE STEP $\alpha$ AND WEIGHT RATIO MULTIPLIER $\beta$

A simple grid search is conducted for $\alpha \in \{0.005, 0.01, 0.02\}$ and $\beta \in \{0.95, 0.9, 0.8\}$ in Fig. 9 for SST-2 and Fig. 10 for MNLI. From the figures, we can find that we are able to trade convergence for efficiency with a more aggressive setting of larger $\alpha$ and smaller $\beta$, yet all results here are decent

with a final accuracy drop of no more than $0.3\%$ for both tasks. Thus, VCAS is robust to different $\alpha$ and $\beta$.

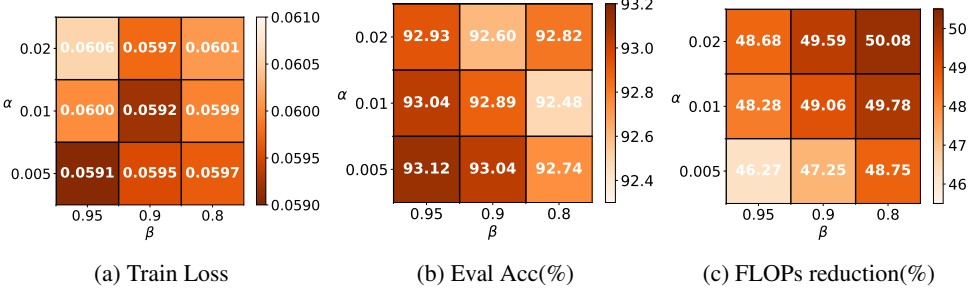

(a) Train Loss                    (b) Eval Acc(%)                    (c) FLOPs reduction(%)

Figure 9: Grid search of $s$ update step $\alpha$ and weight ratio multiplier $\beta$ of BERT-base finetuning on SST-2. The darker color the better.

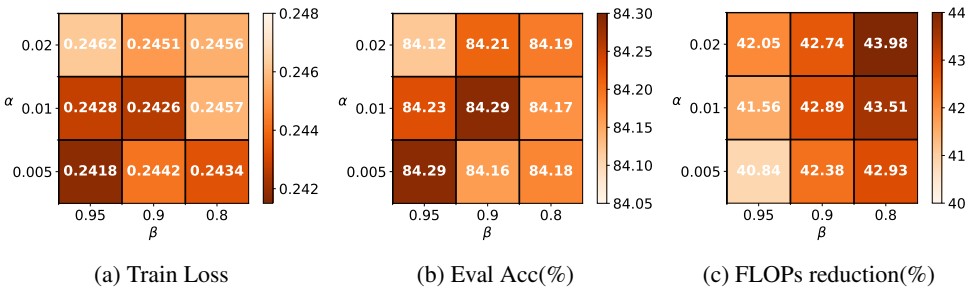

(a) Train Loss                    (b) Eval Acc(%)                    (c) FLOPs reduction(%)

Figure 10: Grid search of $s$ update step $\alpha$ and weight ratio multiplier $\beta$ of BERT-base finetuning on MNLI. The darker color the better.

From all the ablation results above, we can see that VCAS is robust to all these hyperparameters with reasonable values, proving the insensitiveness.

## B  INSIGHTS ON UPDATE OF $s$, $\{\rho_l\}$ AND $\{\nu_l\}$

In this section, we will show how the gradient norm preserving ratio $s$ as well as all the sample ratios $\{\rho_l\}$ and $\{\nu_l\}$ update across the training.

We record the update process of BERT-base finetuning on MNLI with different variance tolerance thresholds $\tau$ as in Appendix. A.1. All results are averaged on three different seeds.

Fig. 11a depicts the update of $s$. For non-decreasing $\{\rho_l\}$, we plot the update of the first and the last values $\rho_1, \rho_L$ in Fig. 11b, with other values lying between. For $\{\nu_l\}$, we show the update of the first three ones $\nu_1, \nu_2, \nu_3$ in Fig. 11c and observe similar behavior of other weights.

It is seen in Fig. 11 that during training of BERT-base on MNLI, the gradient norm preserving ratio $s$ first decreases and then shows a slight downward trend. The activation sample ratios $\{\rho_l\}$ gradually decrease with an abrupt change between epochs due to the rapid decline of train loss caused by the lowered learning rate in the linear learning rate scheduler. The weight sample ratios $\{\nu_l\}$ first decrease and then fluctuate to match the change of activation sample ratios.

## C  PERFORMANCE ON CNN

In Sec. 6, we mainly experiment with Transformer-based models and Adam optimizers. But the variance controlled adaptation depicted in Sec. 5 holds universally for any DNNs with SGD-based optimizers, since it just provides an approximated stochastic gradient with controlled variance to estimate the full gradient. In this section, we employ VCAS on other architectures and other optimizers to prove its versatility.

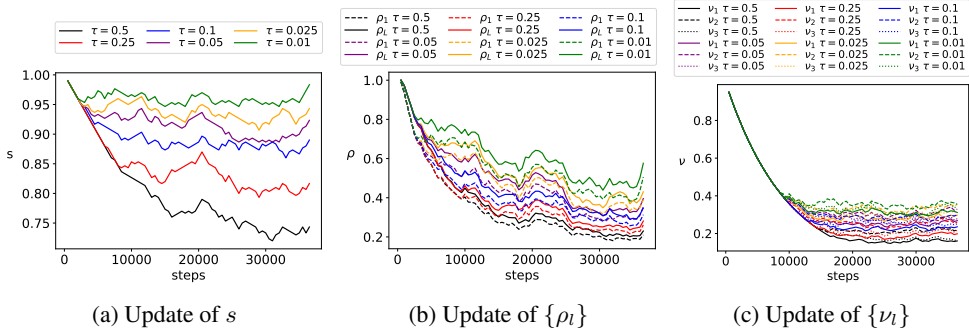

(a) Update of $s$     (b) Update of $\{\rho_l\}$     (c) Update of $\{\nu_l\}$

Figure 11: VCAS update process with different $\tau$ for BERT-base finetuning on MNLI.

For CNN, it is noted that the weight sampler SampleW in Sec. 4 designed for linear layers is not usable for convolution layers. Thus we employ VCAS with a degraded version of activation sampling only.

We experiment with WideResNet-18 with widen factor $w = 4$ pretraining on ImageNet. We use eight NVIDIA 3090Ti to parallel the training with Distributed Data Parallel(DDP). We employ SGDM optimizer with momentum $m = 0.9$. The results are in Tab. 8.

Table 8: Training results of WideResNet-18 pretraining on ImageNet with 8 NVIDIA 3090Ti.

| Method | Train Loss | Eval Acc(%) | Train Time(h) | FLOPs↓(%) | Time↓(%) |
|--------|-----------|-------------|---------------|-----------|----------|
| exact | 1.474 | 75.96 | 21.31 | - | - |
| VCAS | 1.479 | 75.86 | 20.20 | 17.47 | 5.21 |

From the table we can see VCAS is also capable of accelerating the training of CNN. Besides, the parallel setting also proves the parallelizability of VCAS. The relatively low but still decent time reduction can be explained with Amdahl's Law since VCAS only accelerate the calculation part and is not able to accelerate other parts like communication cost during parallel training.

## D  DETAILS ABOUT ALGORITHM. 1

It should be noted that some parts of Alg. 1 are simplified for clarity and we list the implementation details below:

In the algorithm table, we put the calculation of empirical variances out of the two Monte-Carlo loops for simplicity. Yet practically we can calculate $V_{act}$ and $V_w$ inside the loops and average the variance scalars outside. Therefore, we only need to store three tensors additionally regardless of $M$: SGD gradient $G_{s,i}$ to calculate $V_{act}$, and its running mean and running square mean to calculate $V_s$. By sampling only part of parameters to keep gradients, like 1% in our experiments, the memory overhead can be neglected.

Besides, since weight sample ratios $\{\nu_l\}$ are updated parameter-wise according to Eq. 7, the empirical weight variances and SGD variances are also stored parameter-wise when implemented.

Update of activation sample ratios $\{\rho_l\}$ requires finding out gradient sparsity $\{p_l\}$ with the new $s$ according to Eq. 4. In implementation, this is achieved by calculating possible new $\{\rho_l\}$ with both $s + \alpha$ and $s - \alpha$ inside the Monte-Carlo loops and averaging them outside. Then just choose the proper one with new $s$.

# E PROOF

## E.1 PROOF TO UNBIASEDNESS OF VCAS

Let's first consider a $L$-layer MLP. (Note: for simplicity we mildly abuse the term "layer" here, representing a single operation like matrix multiplication and ReLU)

For the last layer $L$, the output gradient $\nabla_{Z^{(L)}}$ is calculated from the loss directly, the same as the Exact BP. Since activation sampler $\hat{\nabla}_{Z^{(L)}} = \text{SampleA}_{\epsilon,\rho_L}(\nabla_{Z^{(L)}})$ is unbiased, we have:

$$\mathbb{E}\left[\hat{\nabla}_{Z^{(L)}}\right] = \nabla_{Z^{(L)}}$$

When back propagation proceeds, we may encounter two types of layers: linear and non-linear. For the linear layer, we have:

$$\nabla_{Z^{(L-1)}} = \hat{\nabla}_{Z^{(L)}} \theta^{(L)}$$

Thus unbiasedness is preserved with the output gradient of the $(L-1)$-th layer:

$$\mathbb{E}\left[\nabla_{Z^{(L-1)}}\right] = \mathbb{E}\left[\hat{\nabla}_{Z^{(L)}}\right]\theta^{(L)} = \nabla_{Z^{(L)}}\theta^{(L)} = \text{Exact BP result}$$

While for the non-linear layer like ReLU, we have:

$$\nabla_{Z^{(L-1)}} = \hat{\nabla}_{Z^{(L)}} \odot J_{Z^{(L)}}$$

where $\odot$ is the Hadamard product and $J_{Z^{(L)}}$ is the Jacobbi matrix determined by $Z^{(L)}$ which is saved in forward pass and is exact. Thus again we derive the the output gradient of the $(L-1)$-th layer being unbiased:

$$\mathbb{E}\left[\nabla_{Z^{(L-1)}}\right] = \mathbb{E}\left[\hat{\nabla}_{Z^{(L)}}\right] \odot J_{Z^{(L)}} = \nabla_{Z^{(L)}} \odot J_{Z^{(L)}} = \text{Exact BP result}$$

Thus by induction, VCAS assures all activation gradients $\hat{\nabla}_{Z^{(l)}}, l = 1 \ldots L$ being unbiased.

Then for weight gradients, since weight sampler $\tilde{\nabla}_{Z^{(l)}} = \text{SampleW}_{\xi_l,\nu_l}\left(\hat{\nabla}_{Z^{(l)}}, Z^{(l-1)}\right)$ is unbiased, we have:

$$\mathbb{E}\left[\tilde{\nabla}_{Z^{(l)}}\right] = \mathbb{E}\left[\hat{\nabla}_{Z^{(l)}}\right] = \nabla_{Z^{(l)}}$$

Finally, we derive all weight gradients being unbiased:

$$\mathbb{E}\left[\nabla_{\theta^{(l)}}\right] = \mathbb{E}\left[\tilde{\nabla}_{Z^{(l)}}\right]^{\top} Z^{(l-1)} = \nabla_{Z^{(l)}}^{\top} Z^{(l-1)} = \text{Exact BP result}$$

For more complicated neural networks like CNN and Transformer, since operations like convolutions and layernorm are all linear transforms, by similar reasoning the unbiasedness still holds.

# F EXPERIMENT DETAILS

## F.1 BERT-BASE PRETRAINING

For BERT-base pretraining we use a crammed BERT in Geiping & Goldstein (2022) with the recipe same as the original settings of 1 day training on a single NVIDIA 2080Ti. The full results are as follows in Tab. 9

From the table we can find that although VCAS achieves a relatively high train loss, the downstream task performance is still competent with exact training. While SB and UB both perform worse on CoLA, which is a vulnerable task, reflecting that they have changed the original convergence trajectory of SGD.

Table 9: Full results on BERT-base pretraining

| Methods | Loss | MNLI-m | MNLI-mm | QQP | QNLI | SST2 | CoLA | STSB | MRPC | RTE | Avg. |
|---|---|---|---|---|---|---|---|---|---|---|---|
| exact | 2.099 | 82.28 | 82.68 | 87.08 | 88.85 | 91.28 | 48.07 | 83.26 | 86.98 | 54.87 | 78.37 |
| SB | 2.133 | 82.34 | 82.86 | 87.27 | 88.63 | 91.28 | 41.82 | 82.86 | 85.53 | 55.23 | 77.53 |
| UB | 2.106 | 82.95 | 83.46 | 87.27 | 88.66 | 91.05 | 42.80 | 83.68 | 85.90 | 55.95 | 77.96 |
| VCAS | 2.134 | 82.03 | 82.82 | 86.92 | 89.23 | 91.62 | 48.36 | 83.02 | 86.03 | 55.23 | 78.36 |

## F.2 RECIPE OF OTHER TASKS

For BERT finetuning, we use AdamW optimizer with $lr = 2e^{-5}$ and $wd = 0.01$. The learning rate scheduler is a linear one with $warmup\_ratio = 0.1$. We set epoch numbers $N = 3$ and a batch size of $batch\_size = 32$.

For ViT finetuning, we use Adam optimizer with $lr = 2e^{-5}$. A linear lr_scheduler with no warmup employed. We run $N = 5$ epochs with batch size $batch\_size = 32$

## G LIMITATIONS

VCAS is designed for adaptively learning the proper sample ratios of large model training on large datasets. It is not suitable for small models with low gradient variances resulting in increased numerical errors, or small datasets with few training steps that is insufficient for the update process in VCAS.

The weight sampler SampleW in VCAS is specially designed for linear layers and is not usable for other operations like convolution. But the activation sampler SampleA can be applied to all mainstream architectures with deep layers. So for CNN or RNN, we need to employ a degraded version of VCAS with activation sampling only, as shown in Appendix. C.

VCAS focuses on mirroring the exact training with theoretical guarantee and is lack of exploration of other possible convergence trajectories that may bring a better result. Thus it is not recommended when the original training recipe is under-optimized.

