# OpenReview forum: "Efficient Backpropagation with Variance Controlled Adaptive Sampling"
_ICLR.cc/2024/Conference — ICLR 2024 poster_

### Official Review · Reviewer_PsQT · 2023-10-30

**Soundness:** 2 fair
**Presentation:** 2 fair
**Contribution:** 2 fair
**Rating:** 6
**Confidence:** 3

**Summary:**

This manuscript proposes a sampling algorithm to eliminate computation during forward and/or BP. More specifically, a variance-controlled adaptive sampling (VCAS) method is designed to accelerate BP by computing unbiased stochastic gradients.

The effectiveness of the VCAS is justified by pre-training and fine-tuning tasks in both vision and NLP domains. Some ablation studies are also included to discuss the effects of hyper-parameters.

**Strengths:**

* This manuscript is well-structured, with a clearly explained methodology section.
* The manuscript evaluates the effectiveness of the VCAS on pre-training and fine-tuning tasks in both vision and NLP domains.
* An ablation study is also included.

**Weaknesses:**

1. Limited literature review. This manuscript did not carefully examine the relevant papers, and some closely related papers were omitted from the discussion, e.g., [1, 2, 3] and the related work in [1].
2. Limited baseline evaluations. Close baseline methods e.g., [2, 3] should be considered.
3. Additional hyper-parameters and insufficient ablation studies. The ablation study on one task, i.e., fine-tuning BERT-base on MNLI, is insufficient to justify the insensitivity of these hyper-parameters.
4. Clarity. Some design choices in Section 5 are just given and have no explanation. For example, how to derive the provided equation from the mentioned zeroth-order algorithm?
5. The claim on the unbiased stochastic gradient needs to be more careful and some theoretical justifications should be provided.

### Reference
1. SparseProp: Efficient Sparse Backpropagation for Faster Training of Neural Networks at the Edge
2. Back Razor: Memory-Efficient Transfer Learning by Self-Sparsified Backpropagation
3. Sparse weight activation training

**Questions:**

NA

---

> ### Author Response · Authors · 2023-11-17
> **Thank you for the valuable feedback!**
>
> We‘d like to thank the reviewer for acknowledging the effectiveness of our work and for the constructive criticism. Below we address reviewer's concerns:
>
> **Q1. Limited literature review on sparsity-based training methods**
>
> Thanks for pointing out relevant works on sparse training. However, we'd like to place our work on another closely related field of ***online batch selection*** that discards less informative data, but not prunes weights or neurons as is typical in sparse training. The main difference comes in two ways:
> - Wall-clock time reduction: Online batch selection methods can always translate FLOPs reduction into time reduction since they neatly wipe the entire datum out. While few sparse training works report time reduction usually for the lack of supportive hardware, especially for unstructured sparsity.
> - Biasedness: In online batch selection, importance sampling is usually adopted which assures unbiasedness. While in sparse training, pruning methods typically cannot guarantee unbiased gradients. Using unbiased gradients for SGD is critical for the algorithm to converge properly.
>
> Moreover, sparse training and online batching selection should be orthogonal since they reduce computation along different dimensions (data vs. model) and the speedup could be multiplicative.
>
> Nevertheless we agree with the reviewer that the paper can be improved with more discussion on sparse training methods. We have added the discussion to Related Work.
>
> **Q2. Limited baseline evaluations**
>
> VCAS can translate FLOPs reduction into time reduction which is rare for sparse training methods. We compare VCAS with the official codebases of methods mentioned by the reviewer:
>
> For Back Razor, we tested on BERT-large finetuning on MNLI. We find though Back Razor can reduce up to 90\% FLOPs, it cannot reduce training time with its unstructured sparsity, and the accuracy drops a lot.
>
> **Tab. Comparison of VCAS with Back Razor**
> |Method|Train Loss|Eval Acc(%)|Train Time(h)|FLOPs Reduction(%)|Time Reduction(%)|
> |-|-|-|-|-|-|
> |Exact|0.1439|86.58|5.48|-|-|
> |Back Razor|0.3827|84.26|5.19|90.00|5.29|
> |VCAS|0.1619|86.63|4.44|44.17|19.03|
>
> SWAT seems close to our work since it conducts a fine-grained pruning on weight and activation and implements a structured sparsity version that uniformly prune channels.
>
> Since the official codebase of SWAT only supports CNNs, we compare SWAT with VCAS on WideResNet18 with widen factor $k=4$ on CIFAR100. We find VCAS can reduce training time while preserving accuracy, while SWAT is even slower than exact (dense) training due to its high overhead.
>
> **Tab. Comparison of VCAS with SWAT**
> |Method|Train Loss|Eval Acc(%)|Train Time(h)|FLOPs Reduction(%)|Time Reduction(%)|
> |-|-|-|-|-|-|
> |Exact|0.0070|73.25|4.32|-|-|
> |SWAT|0.0115|72.01|4.81|50.00|-11.49|
> |VCAS|0.0070|74.16|3.81|19.05|11.72|
>
> **Q3.Additional hyperparameters and insufficient ablations**
>
> First we'd like to emphasize that we employ ***the same set of hyperparameters in all our experiments*** across pretraining and finetuning, CV and NLP, and all get decent FLOPs reduction and accuracy. We believe it can already reflect the insensitivity. As said by Reviewer MWbo in "Strength", "the ablation studies on a few hyperparameters are very important. I see only very few of the papers in this domain have done this before".
>
> From the experimental perspective, we agree with the reviewer that more experiments will help to justify our statement. So we have done ***all ablations again for BERT-base finetuning on SST-2***. Please refer to Appendix A for the results.
>
> **Q4. Clarity of design choices in Section 5**
>
> Sorry for the confusion. To make our algorithm clearer we've added an algorithm table at the end of Section 5. Hope it can relieve your puzzle.
> > How to derive the provided equation from the mentioned zeroth-order algorithm?
>
> The zeroth-order algorithm is embarassingly simple: since the variance increase monotonically with the drop ratio, our algorithm simply decrease the drop ratio by a constant if the variance is too large, and increase the drop ratio if the variance is too small. The only learning signal utilized is whether the variance is higher than the desired threshold. For a more intuitive view of the update, please refer to figures in Appendix C
>
> **Q5. Lack of proof on the unbiasedness of VCAS**
>
> Thank you for posing the vital problem of the unbiasedness of VCAS! We've added our proof in Appendix E.
>
> Proof sketch: since our sampling is unbiased, we just need to prove the unbiasedness preserves in BP. As BP proceeds, linear operations like matmul, convolution and layernorm can obviously preserve unbiasedness. Nonlinear operations like ReLU incur Hadamard product with Jacobian matrix that is determined in forward pass, and thus also preserve unbiasedness. By induction we derive VCAS being unbiased.
>
> We sincerely hope our response can solve your confusion and we would appreciate it pretty much if you could reevaluate our work. Thank you!

---

> > ### Comment · Reviewer_PsQT · 2023-11-19
> > **Response to the authors**
> >
> > The reviewer thanks the authors for providing the feedback.
> >
> > The response has addressed most of the previous concerns. However, before updating the rating, the reviewer would like to invite authors to answer the following questions:
> > * The paper, i.e., Variance Reduction with Sparse Gradients, http://arxiv.org/abs/2001.09623, may achieve a similar effect from a different perspective. It would be great if the authors could comment on this line of research in depth.

---

> > > ### Author Response · Authors · 2023-11-20
> > > **Glad to have addressed most of your concerns!**
> > >
> > > The paper mentioned by the reviewer (hereinafter referred to as *Sparse SVRG*) belongs to the field of SVRG(Stochastic Variance Reduced Gradient) introduced in 2013 by [1], which ***actually cannot “achieve a similar effect with us from a different perspective”*** for the following reasons:
> > >
> > > - Sparse SVRG is ***far more computationally expensive*** than our method. It needs two forward&backward passes in each iteration and additionally calculates the average gradient of a very large batch every few iterations.
> > > - The sparsity introduced by Sparse SVRG is ***unstructured***, which is unable to bring a wall-clock time speedup with current hardwares as is discussed in our earlier reply.
> > > - SVRG methods, including Sparse SVRG, ***fail to outperform SGD*** and can even increase the variance in modern neural networks training as is revealed in [2].
> > > - Our method should be ***orthogonal*** to Sparse SVRG. We can apply our methods to accelerate Sparse SVRG by replacing the SG(stochastic gradient) with ASG(approximated stochastic gradient) calculated with our method.
> > >
> > > Below we provide a detailed comment on this line of research:
> > >
> > > SVRG aims at reducing the variance of SGD for *efficacy*(i.e. lower train loss and higher eval acc with the same epochs) but not *efficiency*(i.e. lower FLOPs and shorter time to achieve a targeted performance) as in our work. In fact, SVRG is much more expensive than SGD in computation: SVRG needs to conduct forward&backward twice in every iteration to get two sets of gradients with respect to current model and a snapshot model. It also needs to compute a full gradient every few iterations by averaging gradients of all data in vanilla SVRG or a very large batch in Sparse SVRG.
> > >
> > > Although SVRG is proven to be better than SGD both theoretically and experimentally in traditional problems like logistic regression, it fails to perform well and even increases the variance of modern neural networks training on harder problems like CIFAR10 as is revealed in [2]. To the best of our knowledge, the first paper to bring the benefit of SVRG into training neural networks at a *practical scale*(i.e. ViT on ImageNet) is [3]  just proposed this year. So till now, SVRG is not widely adopted in practice.
> > >
> > > Sparse SVRG improves upon SVRG by introducing sparsity to gradients to make SVRG "competitive with SGD"(c.f. Conclusion section in Sparse SVRG) in the sense of gradient queries. It is noted that the sparsity is unstructured and cannot bring a wall-clock time reduction with current hardwares as is discussed in our earlier reply.
> > >
> > > Besides, our method should be orthogonal to SVRG methods since we only use ASG(approximated stochastic gradient) in substitute of SG(stochastic gradient) with controlled variance and reduced computation, not relying on the type of optimizer. Variance reduction algorithms in SVRG can still work in combination with our method by replacing SG with ASG that shares similar variance.
> > >
> > >
> > >
> > > **Reference**
> > >
> > > [1] Johnson, Rie, and Tong Zhang. "Accelerating stochastic gradient descent using predictive variance reduction." _Advances in Neural Information Processing Systems_ 26 (2013).
> > >
> > > [2] Defazio, Aaron, and Léon Bottou. "On the ineffectiveness of variance reduced optimization for deep learning." _Advances in Neural Information Processing Systems_ 32 (2019).
> > >
> > > [3] Yin, Yida, et al. "A Coefficient Makes SVRG Effective." _arXiv preprint arXiv:2311.05589_ (2023).

---

> > > > ### Author Response · Authors · 2023-11-22
> > > > **Sincerely looking forward to further discussions**
> > > >
> > > > *Dear Reviewer,*
> > > >
> > > > We hope that our response and revision have adequately addressed your concerns. If our efforts have indeed addressed your concerns, we would be very grateful if you could reconsider our work and possibly adjust the score accordingly. If you have any additional questions or suggestions, we would be happy to have further discussions.
> > > >
> > > > *Best regards,*
> > > >
> > > > *The Authors*

---

> > > > > ### Comment · Reviewer_PsQT · 2023-11-22
> > > > > **Feedback**
> > > > >
> > > > > Thank you for providing the explanation.
> > > > >
> > > > > Though the reviewer might only partially agree with the statements above, he/she is happy to raise the score to 6.

---

> > > > > > ### Author Response · Authors · 2023-11-22
> > > > > > **Thank you for raising the score!**
> > > > > >
> > > > > > Thank you so much for raising the score! We believe the theoretical and experimental success of VCAS can bring a new fashion of variance-controlled sampling to back propagation acceleration. Thank you!

---

### Official Review · Reviewer_Dd1x · 2023-10-30

**Soundness:** 3 good
**Presentation:** 3 good
**Contribution:** 2 fair
**Rating:** 6
**Confidence:** 4

**Summary:**

This work introduces a variance-controlled adaptive sampling (VCAS) method for accelerating the back-propagation of deep neural network training. VCAS computes unbiased, variance controlled gradients for both activations and network weights. By sampling both data samples and tokens in each datum in a layer-wise, fine-grained manner, VCAS can drastically reduce the computation in the back-propagation process without introducing too much variance overhead. With the similar FLOPs reduction, VCAS better optimizes the target model compared with prior loss-based and gradient-based sampling methods.

**Strengths:**

- This work introduces a fine-grained strategy that 1) increasingly removes data samples when back-propagating to the input layer, and 2) samples tokens in each data sample when computing gradients for network weights. This fine-grained strategy allows high FLOPs reduction with controlled variance.

- The sampling ratios are adaptively adjusted according to the estimated variance. As training progresses, the network may require changing sampling ratios, and the adaptive sampling ratios can better satisfy this need.

- Compared with prior methods, the proposed method, VCAS, can better simulate exact back-propagation, leading to better optimized loss and better evaluation performance.

**Weaknesses:**

- Training time reduction: When comparing with baselines, this work uses FLOPs as the efficiency indicator. However, FLOP reduction may not directly translate into wall-clock time reduction due to various factors like parallel computation efficiency. It is suggested to also list the wall-clock training time of each method for a more straightforward comparison.

- Insights on sampling ratio updates: In Section 7 this work has discussed the design choices that determine the sampling ratio updates. For better comprehension, it may be useful to include a figure that shows how $s$ and $\nu_l$ changes as training progresses.

- Figure/Table clarity: Figure 2 seems to lack some more detailed explanation. In Table 1, it is not clear which numbers should be bolded. For example, for ViT-base fine-tuning on CIFAR-100, UB seems to be highlighted for the highest eval accuracy, but for ViT-large fine-tuning on CIFAR-100, UB seems to be highlighted for the lowest train loss? Also for Table 1, how significant is the relative improvement over baselines?

- Limitations: It is suggested to include some detailed discussion on the limitations (e.g., applicable model architectures, base optimizer, dataset) of the proposed method. In this paper, only Transformer-based architectures and Adam-like optimization algorithms are tested. It is not clear whether we can extrapolate the conclusion to other settings.

**Questions:**

- It is not directly clear to me whether the weight gradient sampling of VCAS is applicable to convolution neural networks (CNN). In principle, convolution is yet another linear operator, but I’m not sure how to perform this sampling in a convolutional layer. Similarly, can VCAS be applied when optimizing a recurrent neural network (RNN)?

---

> ### Author Response · Authors · 2023-11-18
> **Thank you for the insightful feedback!**
>
> We thank reviewer Dd1x for acknowledging the effectiveness of our work and for the insightful questions. Below we respond to the questions. We would highly appreciate it if the reviewer agree with our response and consider to raise the score. Thank you so much!
>
> **Q1. Training time reduction.**
>
> Thanks for the suggestion. To compare wall-clock training time, we choose two tasks with the longest training time: BERT-large finetuning on MNLI and ViT-large finetuning on ImageNet. The results are as follows:
>
> **Tab. BERT-large finetuning on MNLI**
> |Method|Train Loss|Eval Acc(%)|Train Time(h)|FLOPs Reduction(%)|Time Reduction(%)|
> |-|-|-|-|-|-|
> |exact|0.1439|86.58|5.478|-|-|
> |SB|0.2492|85.18|4.320|44.44|21.14|
> |UB|0.2266|86.09|4.266|44.44|22.12|
> |VCAS|0.1619|86.63|4.437|44.17|19.00|
>
> **Tab. ViT-large finetuning on ImageNet**
> |Method|Train Loss|Eval Acc(%)|Train Time(h)|FLOPs Reduction(%)|Time Reduction(%)|
> |-|-|-|-|-|-|
> |exact|04135|82.04|52.29|-|-|
> |SB|0.4637|82.21|42.56|44.44|18.61|
> |UB|0.4242|82.21|41.92|44.44|19.83|
> |VCAS|0.4228|82.27|41.28|49.58|21.06|
>
> We find ***VCAS can translate FLOPs reduction into wall-clock time reduction as effectively as simpler online-batch sampling methods*** like UB and SB that drop part of data one-time in a whole, while enjoying mirrored performance with the exact training under theoretical guarantee.
>
> Our current implementation of VCAS is purely based on Python and PyTorch. The gap between FLOPs reduction and time reduction can be further narrowed by optimizing the implementation with CUDA or Triton.
>
> We have added the wall-clock time results to Section 6.2, thanks again for the suggestion!
>
> **Q2. Insights on sampling ratio updates.**
>
> Thanks for the constructive suggestion. We've added a few figures in Appendix B recording the update process of our sample ratios. For all variance thresholds $\tau$ from 0.01 to 0.5, gradient norm preserve ratio $s$ first drops rapidly and then slowly. Activation ratios $\rho_l$ gradually decrease with training. Weight ratios $\nu_l$ first drop and then fluctuate around a certain value that is different across $l$.
>
> **Q3. Figure/Table clarity.**
>
> Sorry for the confusion. Now we've added more explanations in Fig.2. The bold formatting in Tab.1 originally highlights the highest *Eval Acc / (1-FLOPs reduction)* which is a balance between performance and efficiency. Now we've changed it to indicate the best of each metric for better clarity. In Tab.1 we also added underlines to mark Eval Acc that is less than 0.1% off the exact training for a more intuitive look.
>
> **Q4. For Table 1, how significant is the relative improvement over baselines?**
>
> As in Tab.1, the FLOPs improvement is relatively marginal compared with UB and SB in our *conservative settings* of variance tolerance ratio $\tau=0.025$. But the ***accuracy improvement is significant***, where the train loss and eval accuracy of VCAS are much closer to exact training.
>
> However, it should be noted that the usage of VCAS is different from existing sampling baselines like UB and SB. For UB and SB, we need to preset a sample ratio $r$ carefully and pray for a decent outcome. While for VCAS, we just need to set a variance tolerance ratio $\tau$ far less than 1, and then it can adaptive learn the proper fine-grained sample ratios which can keep the training curve closely to that of exact training. Therefore, two more important improvements are that ***VCAS is tuning-free for sample ratios*** and ***the FLOPs reduction achieved by VCAS is "safer" with theoretical guarantee*** compared to other baselines.
>
> **Q5. Limitations should be discussed.**
>
> Thanks for the constructive advice. We've added a detailed discussion of the limitations of VCAS in Appendix G.
>
> Regarding to the limitations on applicable model architecture, base optimizer, dataset, theoretically, there will not be such limitations. But the performance of VCAS is closely related with the "redundancy" of architectures and datasets. Nevertheless, in such scenario, VCAS can learn to keep all data to control the variance (i.e., fall back to exact training), and the accuracy will not be harmed. On the other hand, other online batch sampling methods are likely to fail.

---

> > ### Author Response · Authors · 2023-11-18
> > **Part 2**
> >
> > **Q6. Whether can VCAS apply to other architectures?**
> >
> > Yes. We apply VCAS to train WideResNet-18 on ImageNet with Momentum SGD, i.e., we generalize it beyond Transformers and beyond Adam. For CNNs, the weight gradient sampling part is not applicable, since the "tokens" (i.e., "pixels") in CNNs are not individually processed, but tightly coupled. Therefore, we only apply activation gradient sampling and adapataion of VCAS.
> >
> > We've conducted an experiment using 8\*3090Ti to pretrain WideResNet-18 with widen factor $k=4$ on ImageNet. We use SGDM with $momentum=0.9$. The results are as follows:
> >
> > **Table. VCAS performance for training WideResNet-18 on ImageNet**
> > |Method|Train Loss|Eval Acc(%)|Train Time(h)|FLOPs Reduction(%)|Time Reduction(%)|
> > |-|-|-|-|-|-|
> > |exact|1.474|75.96|21.31|-|-|
> > |VCAS |1.479|75.86|20.20|17.47|5.21|
> >
> > From the table we can see that VCAS is also capable of ***CNN acceleration***, and also usable for ***other optimizers like SGDM*** here. Besides, the parallel setting proves the ***parallelizability*** of VCAS.
> >
> > Results above are added to Appendix C.

---

> > > ### Comment · Reviewer_Dd1x · 2023-11-19
> > > **Reviewer's Response**
> > >
> > > The authors' detailed responses have addressed most of my previous concerns. Thank you for providing the helpful explanation and additional analysis. I'm now leaning towards accepting this work.

---

> > > > ### Author Response · Authors · 2023-11-19
> > > > **Thank you for raising the score!**
> > > >
> > > > Thank you so much for raising the score! We believe the theoretical and experimental success of VCAS can bring a new fashion of variance-controlled sampling to back propagation acceleration. Thank you!

---

### Official Review · Reviewer_d6vC · 2023-10-31

**Soundness:** 4 excellent
**Presentation:** 3 good
**Contribution:** 3 good
**Rating:** 6
**Confidence:** 3

**Summary:**

This paper proposes Variance-Controlled Adaptive Sampling (VCAS), which performs an approximated stochastic gradient with an adaptive sampling rate. Based on the insight that gradients are sparse after learning has progressed to some extent, the authors improve the efficiency of learning by computing only a few selected gradients through adaptive sampling. The proposed method approximates the exact backpropagation values well in BERT and ViT training.

**Strengths:**

VCAS performs backpropagation 20-50% faster, while following the loss curve of true full backpropagation with low variance. VCAS has an adaptive sampling rate, which allows for efficient sample selection based on learning loss and per layer. The idea is simple and highly applicable.

**Weaknesses:**

Comparing accuracy under the same amount of FLOPs reduction makes it difficult to understand its effectiveness compared to a metric like time to reach target accuracy[1]. As a result, it is unknown how VCAS will perform under a 50% or greater reduction.


[1] Mindermann, S., Brauner, J. M., Razzak, M. T., Sharma, M., Kirsch, A., Xu, W., ... & Gal, Y. (2022, June). Prioritized training on points that are learnable, worth learning, and not yet learnt. In International Conference on Machine Learning (pp. 15630-15649). PMLR.

**Questions:**

I would like to see more detail in Table 2 or 3. What is the relationship between Final Eval Accuracy and FLOPs reduction? For example, is the recommending FLOPs reduction ratio for VCAS around 40%?

---

> ### Author Response · Authors · 2023-11-18
> **Thank you for the valuable feedback!**
>
> We thank reviewer d6vC for assessing our method as ***simple and highly applicable*** and for the valuable questions. Below we respond to the questions.
>
> **Q1. Relationship between Eval Acc and FLOPs reduction.**
>
> Thank you for posing the question that strikes at the heart. Actually that's where VCAS differs from previous sampling methods in essence.
>
> In previous sampling methods, we typically first determine a sample ratio(FLOPs reduction ratio) $r$ and then apply sampling algorithms with it. However, many recent works\[1,2\] that reflect the efficient training community have pointed out that this pipeline is indeed unfairly trading accuracy for efficiency. The sample ratio $r$ actually contains prior knowledge for the incoming training by telling the model how much percentage of the dataset is less informative.
>
> VCAS distinguishes from previous sampling methods in that it eliminates the need of determining sample ratios. With VCAS, we use the variance thresholds to determine proper sample ratios adaptively. And to our best knowledge, ***we are the first to conduct adaptive unbiased sampling in training while preserving performance under theoretical guarantee***.
>
> Now we can answer your question: There is ***no direct relation*** between Eval Acc and FLOPs reduction in VCAS. They are ***connected with variance thresholds*** $\tau$ in the following way: FLOPs reduction will consistently climbs up as $\tau$ goes larger since we can tolerate a more radical sampling scheme. While for Eval Acc, when $\tau$ is not that large, like 0.025 in our setting, it won't change too much compared with exact training. But as $\tau$ goes larger, the extra sampling variance becomes unignorable, thus the convergence is affected and Eval Acc will drop.
>
> **Q2. More detail in Table 2 or 3.**
>
> For Table 2 now moved to Table 5, we are varying variance threshold $\tau$ . The FLOPs reduction consistently increases with bigger $\tau$. The Eval Acc keeps within 0.1% fluctuation of exact result when $\tau\le0.05$ which means the extra variance is no more than 10% and can be neglected, but drops a lot when $\tau\ge0.1$ which means more than 20% variance is introduced that affects convergence.
>
> For Table 3 now moved to Table 7, we are varying update frequency $F$. As $F$ increases, we will conduct fewer updates, so the FLOPs reduction first increases due to reduced overhead caused by update process, and then decreases due to insufficient number of updates to converge on adequate sample ratios. The Eval Acc shows no much difference across $F$ since it's theoretically guaranteed to produce a similar result with the exact training with a low $\tau$, like $\tau=0.025$ here.
>
> **Q3. Is the recommending FLOPs reduction ratio for VCAS around 40%? How VCAS will perform under a 50% or greater reduction?**
>
> As discussed above, the FLOPs reduction of VCAS is not tuned manually but learned in the training process. So ***the FLOPs reduction ratio of VCAS varies*** among models and datasets and can be learned automatically, ***reflecting how much FLOPs can be "safely" saved under given variance budget***.
>
> Indeed, we can still trade accuracy for efficiency with a much larger variance in VCAS. For example, we test with BERT-large finetuning on MNLI with a big $\tau=0.5$ that provides a 50% FLOPs reduction, which means introducing *100% extra variance* actually. For comparison we also tested SB and UB with FLOPs reduction of 50%. The results are as follows:
>
> **Table. Different methods with FLOPs reduction of 50%, strikethrough indicates divergence**
>
> |Method|Train loss|Eval Acc(%)|FLOPs reduction(%)|
> |-|-|-|-|
> |exact|0.1439|86.58|-|
> |SB|~~1.099~~|~~32.74~~|50.00|
> |UB|~~1.104~~|~~32.74~~|50.00|
> |VCAS|0.2633|86.01|50.05|
>
> We can see that the performance of VCAS drops a lot compared to exact training, which is actually predicted by the large $\tau$ value. However, the performance of VCAS is still much better than baselines like SB and UB both of which do not converge under this setting, owing to the fine-grained sampling and adaptive sample ratios across different training phases.
>
> **Q4.  Effectiveness of time to reach target accuracy.**
>
> We list the effectiveness of time to reach target accuracy for BERT-base finetuning on MNLI below:  (exact training results in Final Eval Acc of 84.33 and total steps of 36816)
>
> **Table. Number of steps to reach target accuracy with different methods**
>
> |Target Acc(%)|exact|SB|UB|VCAS|
> |-|-|-|-|-|
> |75|1.5k|3.5k|1.5k|**1.5k**|
> |80|4.5k|9.5k|5.5k|**4.5k**|
> |84|20k|not reached|not reached|**19k**|
>
> We can find VCAS similar with exact and much better than SB and UB.
>
> For more insights on the effectiveness of VCAS, please refer to Fig.6b where we presented the Eval Acc curve of these methods.
>
> **Reference**
>
> [1]"The Efficiency Misnomer." ICLR.2021.
>
> [2]"No train no gain: Revisiting efficient training algorithms for transformer-based language models." arXiv.2023.

---

> > ### Author Response · Authors · 2023-11-21
> > **Sincerely looking forward to further discussions**
> >
> > *Dear Reviewer,*
> >
> > We hope that our response and revision have adequately addressed your concerns. If our efforts have indeed addressed your concerns, we would be very grateful if you could reconsider our work and possibly adjust the score accordingly. If you have any additional questions or suggestions, we would be happy to have further discussions.
> >
> > *Best regards,*
> >
> > *The Authors*

---

### Official Review · Reviewer_MWbo · 2023-11-06

**Soundness:** 3 good
**Presentation:** 3 good
**Contribution:** 3 good
**Rating:** 8
**Confidence:** 4

**Summary:**

The authors propose a sampling method for back propagation with controlled variance and self-adaptive sample ratios, named VCAS. It computes an approximate stochastic gradient by applying finegrained sampling to gradually remove samples and tokens during backpropagation. VCAS have similar variance as well accuracy with exact back propagation, while seems to reduce the training cost significantly.

**Strengths:**

1. I like it very much that the authors test the proposed algorithm on multiple fine-tuning and pre-training tasks in both vision and
natural language domains. Typically, papers in this domain would use small scale datasets such as MNIST or CIFAR 10/100.

2. The ablation studies on a few hyperparameters are very important. I see only very few of the papers in this domain have done this before.

**Weaknesses:**

I think the authors can improve the paper in the following ways:

1. I believe adding an algorithm table with detailed steps would make the paper more clear.

2. The authors report the Final Train Loss / Final Eval Acc.(%) / FLOPs reduction ratio(%). However, I'd like to know the actual reduction in training time as these sampling methods might introduce overhead in computation. It would be helpful if the authors can report a time table for training on these datasets.

To be frank, I feel not many papers actually do this but it can be interesting to see that the actual training time might not be reduced at all, or at least not much as expected given a certain sampling ratio.

3. The paper lacks discussions of related paper. For example, https://arxiv.org/pdf/2104.13114.pdf also considers the importance sampling problem by sampling data points proportionally to the loss, instead of norm of gradient.

For another example, https://arxiv.org/pdf/2306.10728.pdf also proposes adaptively sampling methods for dynamically selecting data points for mini-batch. I'd love to see the authors discussed more about these papers.

4. Can the authors be more specific in terms of the notations? Adding a table of notation would be very helpful. For example, what is $h^{(l)}$ below:

$$\nabla_{Z^{(l-1)}}=h^{(l)}\left(\nabla_{Z^{(l)}} ; Z^{(l-1)}, \theta^{(l)}\right)$$

**Questions:**

1. Although the proposed VCAS algorithm seems promising compared with SB and UB, I'd like to know the actual reduction in training time as these sampling methods might introduce overhead in computation. It would be helpful if the authors can report a time table for training on these datasets.

---

> ### Author Response · Authors · 2023-11-18
> **Thank you for the insightful feedback!**
>
> We thank reviewer MWbo for acknowledging the effectiveness of our work and for the constructive questions. Below we respond to the questions.
>
> **Q1. Add an algorithm table.**
>
> We appreciate it a lot for your constructive advice. We've now added an algorithm table in Algorithm 1 at the end of Section 5. Thank you!
>
> **Q2. Report wall-clock time reduction.**
>
> Thanks for your insightful advice. As you can see, not many papers report the wall-clock time reduction, mainly because the wall-clock time is highly dependent on hardware and implementation. FLOPs reduction, however, is a high-level metric directly reflecting the real power of algorithm. Each metric has its own pros and cons.
>
> Though, we agree that wall-clock time is indispensable since that's what we really want to accelerate and FLOPs reduction can not always translate into equal wall-clock time reduction, not only because of the overhead introduced, but also due to parallelizability, hardware friendliness, and compatibility to many other minor but crucial parts in the training procedure.
>
> For this metric we choose two tasks with the longest training time, which are most worthy of acceleration: BERT-large finetuning on MNLI and ViT-large finetuning on ImageNet. The results are as follows:
>
> **Tab. BERT-large finetuning on MNLI:**
>
> |Method|Train Loss|Eval Acc(%)|Train Time(h)|FLOPs reduction(%)|Time reduction(%)|
> |-|-|-|-|-|-|
> |exact|0.1439|86.58|5.478|-|-|
> |SB|0.2492|85.18|4.320|44.44|21.14|
> |UB|0.2266|86.09|4.266|44.44|22.12|
> |VCAS|0.1619|86.63|4.437|44.17|19.00|
>
> **Tab. ViT-large finetuning on ImageNet:**
>
> |Method|Train Loss|Eval Acc(%)|Train Time(h)|FLOPs reduction(%)|Time reduction(%)|
> |-|-|-|-|-|-|
> |exact|04135|82.04|52.29|-|-|
> |SB|0.4637|82.21|42.56|44.44|18.61|
> |UB|0.4242|82.21|41.92|44.44|19.83|
> |VCAS|0.4228|82.27|41.28|49.58|21.06|
>
> From these tables, we can find that VCAS can translate FLOPs reduction into wall-clock time reduction ***as effectively as simpler online-batch sampling methods*** like UB and SB that drop part of data one-time in a whole, while enjoying mirrored performance with the exact training under theoretical guarantee.
>
> Our current implementation of VCAS is purely based on Python and PyTorch. The gap between FLOPs reduction and time reduction can be further narrowed by optimizing the implementation with CUDA or Triton.
>
> We have added the wall-clock time results to Section 6.2, thanks again for the suggestion!
>
> **Q3. Lack discussions of related paper.**
>
> Thanks for your advice! We have added more discussions on these papers as well as many other papers in Section 2 of Related Work in our paper.
>
> Overall, we are addressing the problem of sampling in a new way to the best of our knowledge. ***VCAS differs from all these sampling algorithms in that it doesn't need a carefully tuned sample ratio and is always sure to provide a similar result with exact training under theoretical guarantee***.
>
> Specifically for the two methods mentioned by the reviewer, the loss based method OBFTF samples data by minimizing the distance of average losses between the whole batch and the sampled batch. Its theoretical guarantee and stability are not strong enough since it is sensitive to the absolute value of losses. It is also sensitive to batch size for solving a combinatorial optimization problem that will explode when batch size is large.
>
> The adaptive method AdaSelection is an ensemble of SB, UB, uniform sampling and other methods by adaptively assigning weights among them. It is different from VCAS in the way of adaption: VCAS adaptively find the proper sample ratios, while AdaSelection still needs to preset a sample ratio and simply adapt weights of each method under the ratio. Thus VCAS is more tuning-free, more effective with finer sampling granularity, and provides more theoretical guarantee.
>
> **Q4. Clarity of notations.**
>
> Sorry for the confusion. $h^{(l)}$ means the function of calculating input gradient with output gradient, input and weight. For example for linear layers, we have input gradient equals to output gradient multiplied by weight. The same with $g^{(l)}$ for weight gradient.
>
> We have now added more explanations on every notations in Section 4 and 5. Hope it can relieve your puzzle!

---

> > ### Author Response · Authors · 2023-11-21
> > **Sincerely looking forward to further discussions**
> >
> > *Dear Reviewer,*
> >
> > We hope that our response and revision have adequately addressed your concerns. If our efforts have indeed addressed your concerns, we would be very grateful if you could reconsider our work and possibly adjust the score accordingly. If you have any additional questions or suggestions, we would be happy to have further discussions.
> >
> > *Best regards,*
> >
> > *The Authors*

---

> > > ### Comment · Reviewer_MWbo · 2023-11-22
> > >
> > > thank you for your prompt and detailed responses! the authors addressed my questions well, especially for the training time reduction part!
> > >
> > > raise my score

---

> > > > ### Author Response · Authors · 2023-11-22
> > > > **Thank you for raising the score!**
> > > >
> > > > Thank you so much for raising the score! We believe the theoretical and experimental success of VCAS can bring a new fashion of variance-controlled sampling to back propagation acceleration. Thank you!

---

### Meta-Review · Area_Chair_oS7V · 2023-12-06

**Metareview:**

In extensive experiments, the paper shows that further approximations of the stochastic gradient can significantly speed up training of deep nets without a loss in accuracy. Specifically, a method called 'Variance Controlled Adaptive Sampling' (VCAS) is proposed which subsamples the weight- and activation-gradients appearing in backpropagation.

The main strength of the paper are the extensive experiments running vision-transformers and language models on large datasets. These go beyond the usual CIFAR-10/CIFAR-100 datasets used in related works, and demonstrating an improved performance in this scale is a significant contribution and of interest to the community.

The main weaknesses (as pointed out by the reviewers) are the clarity (figures, tables, reasoning behind the design-choices) and the literature review / comparison to baseline methods. The proposed method appears to work best for training large transformers on large datasets (which is highly relevant), but for example the ImageNet results are perhaps less convincing.

Overall, the paper makes a solid contribution to improving the training speed of large neural networks. I recommend to accept this paper as a poster.

**Justification For Why Not Higher Score:**

The paper builds mostly on well-known techniques and is in some sense a bit incremental.  It is unclear to me at this stage whether the paper can serve as a basis for future innovations, or provides deep insights into the training speed of deep neural networks. For an oral or spotlight recommendation, I would have like to see some of these.

**Justification For Why Not Lower Score:**

The paper is a solid work for improving the training speed of neural networks, and will be of use for the community, especially if the code is released. All reviewers recommended acceptance, and acknowledged the strength of the experiments. Most concerns of the reviewers were addressed in the rebuttal.

---

### Decision · Program_Chairs · 2024-01-16

Accept (poster)